# Will We Have a Cohort of Healthcare Workers Full Vaccinated against Measles, Mumps, and Rubella?

**DOI:** 10.3390/vaccines8010104

**Published:** 2020-02-27

**Authors:** Andrea Trevisan, Chiara Bertoncello, Elisa Artuso, Clara Frasson, Laura Lago, Davide De Nuzzo, Annamaria Nicolli, Stefano Maso

**Affiliations:** Department of Cardiac Thoracic Vascular Sciences and Public Health, University of Padova, 35128 Padova, Italy; chiara.bertoncello@unipd.it (C.B.); elisa.artuso@unipd.it (E.A.); clara.frasson@unipd.it (C.F.); laura.lago@unipd.it (L.L.); ddenuzzo@gmail.com (D.D.N.); annamaria.nicolli@unipd.it (A.N.); stefano.maso@unipd.it (S.M.)

**Keywords:** rubella, mumps, measles, healthcare workers, immune coverage, vaccination coverage

## Abstract

Healthcare workers are a population exposed to several infectious diseases, and an immunization programme is essential for the maintenance of good vaccination coverage to protect workers and patients. A population of 10,653 students attending degree courses at Padua Medical School (medicine and surgery, dentistry and health professions) was screened for vaccination coverage and antibody titres against rubella, mumps, and measles. The students were subdivided into five age classes according to their date of birth: those born before 1980, between 1980 and 1985, between 1986 and 1990, between 1991 and 1995, and after 1995. Vaccination coverage was very low in students born before 1980, but the rate of positive antibody titre was high due to infection in infancy. Increasing date of birth showed increased vaccination coverage. In contrast, immune coverage was high for rubella (more than 90%) but not for mumps and measles (approximately 80%). An “anomaly” was observed for mumps and measles in the cohort born between 1991 and 1995, probably due to the trivalent vaccine formulation. Students born after 1990 showed vaccination coverage that exceeded 90%. It is therefore very likely that we will have a future generation of healthcare workers with optimal vaccination coverage.

## 1. Introduction

Healthcare workers (HCWs) constitute a population potentially exposed to transmittable infectious disease because of their work. Owing to this potential exposure, the Italian “National Vaccination Prevention Plan” 2017–2019 strongly suggests that healthcare workers (HCWs) be vaccinated against seven transmittable diseases. Furthermore, in 2017, the so-called “Pisa card” was drawn up by several scientific societies during the national conference “*Medice cura te ipsum*” (Pisa 27–28 March 2017). However, currently, there are no mandatory vaccinations for HCWs in Italy.

Among the seven suggested vaccines are those against rubella (Ru), mumps (Mu) and measles (Me), usually combined as a trivalent vaccine (MMR), introduced in Italy in 1999 (circular n. 12) and approved by the National Plan for the Eradication of Measles and Congenital Rubella (Ministry Decree November 29, 2001), according to the objectives of the World Health Organization (WHO, Global Measles and Rubella Strategic Plan 2012–2020).

The MMR vaccination schedule provides a first dose in the second year of life and a second dose at 5–6 (recommended) or 11–12 years of age. As a consequence, the incidence of Me dramatically decreased over time. In contrast, probably due to, on the one hand, a “loss of confidence” with the disease and, on the other hand, the growth of the no-vax movements, the trend in vaccine coverage (in particular for MMR) in Italy decreased over time [1]. This decrease was most likely the main cause of the Me outbreak in 2017. As a consequence, since June 7, 2017, MMR is mandatory in Italy (Law Decree n. 73) for new-borns and for adolescents until 16 years of age.

A two-dose schedule is universally recommended for Me [2], with an effectiveness of more than 95% after the first dose and an increase to 99% after the second dose. Immunity persists for a long time [3]. On the other hand, as ascertained previously, Ru protection generally appears to be high, whereas Mu and Me protection does not [4].

The objectives of the present research, performed in a large cohort of students belonging to the Medical School of Padua University susceptible to contract and/or to transmit the infection owing to their future job, were analysed for the MMR vaccine and immune coverage with the aim to evaluate whether, in the near future, it will be possible to have a new generation of HCWs fully vaccinated (and immunized) against Me, Mu, and Ru. Answers were also sought to other questions that arose during the analysis of the results.

## 2. Materials and Methods 

### 2.1. Population 

According to Italian law on occupational safety and health (legislative decree 81/08), students attending to the degree courses of the Medical School of Padua University (medicine and surgery, dentistry, and health professions) are subjected to health surveillance in the second (dentistry and medicine and surgery until 2017) or the first (medicine and surgery from 2018 and health professions) year of coursework. From 2004 until 2019, students were screened for antibodies against transmissible but preventable diseases, including Me, Mu, and Ru. The enrolment conditions were as follows: the students were born in Italy (thus, they had the same vaccination schedules); and their certificate of vaccination, released from the Public Health Office, stating whether the student had been vaccinated against Ru, Mu, Me or MMR, was available. In total, 10,653 students were enrolled (3714 males and 6939 females, male/female ratio 0.54).

The enrolled students included 4705 students (44.2%) from medicine and surgery, 345 students (3.2%) from dentistry, and 5603 students (52.6%) from health professions. They were mostly from Northern Italy (93.8%), and among these, 86.1% were from the Veneto region (86.1%), predominantly from the province of Padua (33.9%).

The students were further subdivided according to the year of birth, from 1980 to 2000, and were categorized into classes as follows: those born between 1980 and 1985, 1986 and 1990, 1991 and 1995, and after 1995. Students born before 1980 were included in one class.

This was an observational study in which we analysed data from a mandatory health surveillance activity on workers exposed to biological risks regulated by the Italian law D. Lgs. 81/2008; consequently, evaluation by an ethics committee was not necessary. However, students signed an informed consent form on the processing of personal and sensitive data, in which they also expressed consent to the possibility that the data collected would be processed anonymously for epidemiological investigations and/or for scientific research purposes.

### 2.2. Measurement of MMR Antibodies

The measurement of MMR IgG antibodies was carried out by means of the commercial enzyme-linked immunosorbent assay (ELISA) Enzygnost (Dade Behring, Marburg, Germany). Antibody levels of Ru and Me were reported as positive (higher than 10 or 350 IU/L, respectively), negative (lower than 5 or 150 IU/L, respectively) or equivocal (5–10 or 150–350 IU/L, respectively). Mu antibody measurement was quantitative. According to the Center for Disease Control and Prevention (CDC) recommendations [5], equivocal results were processed as negative.

### 2.3. Statistics

A 2 by 2 chi-squared (χ^2^) test (Yates correction) was the statistical method used to compare the seroprevalence of antibodies. Other statistical analyses were descriptive. Further, multiple linear regression was used to analyse the variables influencing antibody behaviour, and the following outcomes were considered for the purpose of multiple linear regression if appropriate (independent variables): (1) the age at first dose of vaccine, (2) the date of the first dose of vaccine, (3) the date of birth, and (4) the interval since the dose (or the last dose) of vaccine; antibody titre (see below) was the dependent variable. For better interpretation of the results, antibody titres were categorized as 1 if negative, 2 if equivocal, and 3 if positive. Significance was defined by *p* < 0.05. Statsdirect version 2.7.7 (Statsdirect Ltd., UK) was used for statistical analyses.

## 3. Results

A high percentage of students born before 1980 (Figure 1, panel A) were not vaccinated against Ru, Mu or Me (74.1%, 97.4%, and 87.7%, respectively). Among those not vaccinated against Ru, females showed better compliance (“only” 56.5% not vaccinated) than males (95.5% not vaccinated). This is because vaccination against Ru was actively offered during female adolescence at the end of primary school or at the beginning of secondary school from 1972 to 1988–1989.

Furthermore (data not shown), among the subdivisions of the date of birth, in those born before 1960 (13 subjects), between 1960 and 1969 (73 subjects), and between 1970 and 1979 (409 subjects), the percentage of individuals not vaccinated against exanthematic diseases was 100% in the first group; 94.5% (Ru), 98.6% (Mu) and 97.3% (Me) in the second group; and 69.7% (Ru), 97.1% (Mu) and 85.6% (Me) in the third group. Despite non-vaccination, the cohort born before 1980 showed a high percentage (higher than 90%, except Mu, 77%) of positive antibodies (Figure 2, panel A).

With an increasing date of birth, the number of non-vaccinated subjects decreased progressively to 5.2% if the students were born after 1990 and to 2.9% if born after 1995 (Figure 1, panel A). Among subjects born between 1980 and 1990, the number of vaccinated subjects increased, with one dose (cohort 1980–1985) and two doses (cohort 1985–1990) (Figure 1, panels B and C, respectively). Overall (Figure 1, panel D), more than 90% of the cohorts born after 1990 had at least one dose of MMR vaccine, and most had two doses; if they were born after 1995, the vaccination coverage exceeded 95% (at least one dose) and reached 90% with two doses.

From what was reported in vaccination certificates, several combinations of Ru, Mu, or Me vaccines were administered (Table 1). Children born before 1980 and during the eighties were prevalently vaccinated (when vaccinated) with a single dose of Ru, Mu, or Me alone; the implementation of the MMR formulation (one dose only) started in the eighties, with a peak in 1986, and decreased throughout the years. A consistent cohort of children born after 1989 began to be vaccinated with two MMR doses (54.8% of subjects), with a progressive increase during the nineties, with almost 95% coverage in children born in 1998 (Figure 1, panel A–D). As indicated in the box at the bottom of panel D of Figure 1, vaccination coverage with at least one dose, starting from those born in 1991, was greater than 94% and reached 97.8%–100% in those born in 1999 and 2000.

Before the implementation and the achievement of high compliance with the MMR vaccine, the behaviour of parents regarding childhood vaccinations was different. Ru vaccination (Figure 1, panel C) was highly administered in more than 70% of females (commonly one dose). Generally, compliance with the Mu vaccine alone was low (Figure 1, panel B), at least until the implementation of the MMR formulation. In contrast, compliance with the Me vaccine (Figure 1, panel B–D), usually with one dose of Me alone, was sufficiently high (more than 50%) by the beginning of the eighties.

The positivity of antibodies to Ru, Mu, and Me was high in older non-vaccinated students, but this rapidly decreased with an increasing date of birth (Figure 2, panel A). On the other hand, after vaccination, a different behaviour was observed on the basis of the number of doses and the vaccination type (Figure 2, panel B–D). One or two doses of the Ru vaccine induced a high antibody response of over 90%.

An “anomaly” immediately catches the eye, i.e., the positive response to vaccination against Mu (1 dose) and Me (1 and 2 doses) in the age group born between 1991 and 1995, with the number of positive subjects being significantly lower (*p* < 0.0001) than that in the other age classes. To understand this "anomaly”, the year of birth (Figure 3, panel A,B) and the year of the first vaccine dose for Mu and Me (Figure 3, panel C–D) were analysed separately, also analysing the three years before 1991 and the two years after 1995. The “indicted” years are 1991–1994 (Me one dose), 1991–1993 (Me two doses) and 1992–1995 (Mu one dose) according to the date of birth and 1993–1995 (all types of vaccine) according to the date of the first dose of vaccine. The second dose of vaccine against Mu, but only partially against Me (independently by the years of vaccination), largely increased the percentage of positive subjects. A multivariate analysis (data not shown) applied to this cohort confirmed that the date of birth and the date at the first dose of vaccine significantly (*p* < 0.0001) influenced antibody titre after 1 dose of Mu vaccine and 1 or 2 doses of Me vaccine, but not after 2 doses of Mu vaccine.

In the whole population, the multivariate analysis (Table 2 and Table 3) showed that date of birth influenced antibody titre only if subjects were vaccinated once (Ru, Mu, and Me), whereas the date of vaccine or the 1st dose of vaccine influenced Ru (once) or Me (twice) and the interval since the vaccine (or 2nd dose) influenced Ru and Me (once and twice), but not Mu.

Me Ab one dose = 5.806215 −0.000058 date of birth −0.000027 date of vaccine −0.000054 interval.

Table 4 and Table 5 show the average ages to which the only dose of vaccine (vaccinated once) or the first and second dose (vaccinated twice) were administered, respectively. 

## 4. Discussion

The aim of this research was based on two issues: (1) vaccination and (2) immune coverage. Vaccination and immune coverage do not always go together, and the reason for this discrepancy can be attributed primarily to five causes: (1) the type of vaccine used, (2) the interval since the last dose, (3) age [6], (4) sex differences [7], and (5) racial differences [8,9].

The analysis of data on vaccination and immune coverage in a cohort of medical students shows that date of birth and the date of the first dose of vaccine widely influence immune response. In addition, there are two further lines of evidence, namely, (1) the progressive awareness of parents about the need to vaccinate their children against Me, Mu, and Ru and (2) probable differences in the formulation of vaccines.

According to the second hypothesis, certainly the most striking is the introduction of the Rubini strain against Mu in 1991 in the TRIVIRATEN^®^ formulation, which was proven to be ineffective and was subsequently withdrawn from the market. A comparative study between the different strains in use against Mu, i.e., Jeryl-Lynn, Urabe, and Rubini, showed that the efficacy was 80.7%, 54.4% and -55.3%, respectively [10]. This may explain the response to vaccination in the 1991–1995 cohort, which was significantly lower after one MMR dose. On the other hand, the same cohort vaccinated twice reached an immune coverage of 80% or more; two doses of vaccine were already necessary to maintain immune coverage.

In the same cohort, the significant decrease in immune coverage after vaccination against Me appears less explainable, showing a similar trend to that of Mu, but not corrected by the second dose. MORUPAR^®^ (Schwartz strain) was also introduced in those years and was then withdrawn in early 2000, but not because of the ineffectiveness of the strain but because of problems concerning a higher incidence of serious allergic reactions. Currently, we have no convincing explanations for this phenomenon except a possible "negative" influence of the Rubini strain on the Edmonton–Zagreb strain used for Me, but it remains inexplicable why the second dose, as in the case of Mu, did not remedy the failure of the first. Unfortunately, the type of vaccine used is not information that is provided in vaccination certificates. In addition, in a comparison of this phenomenon between years of first dose of vaccine and the date of birth, the loss in immune coverage is delayed by 1–1.5 years with respect to the date of birth because vaccination is usually administered after the first year of age. There appears to be some hesitancy in children vaccinated with a single dose, even for those born in more recent years. This hesitancy could explain the non-completion of the vaccination cycle.

The necessity of vaccination implementation is related to the decline in subjects naturally immunized over time. It is known that the circulation of Ru, Mu and, above all, Me is dramatically decreased compared to 40 years ago [11], but Me virus after infection may cause subacute sclerosing panencephalitis [12,13] or, when infecting immune cells, acute immune suppression with a reduction in humoral immune system and a low response to future infectious diseases [14,15]. These effects are not observed after vaccination.

A relevant question is that, despite good vaccination coverage, the immune coverage for Mu and Me is, on average and minus the problems concerning the 1991-1995 cohort, lower than 90%; on the contrary, immune coverage for Ru is higher than 95%, even after only one dose of vaccine. For this reason, for Mu and Me herd immunity, which are over 90% and over 95%, respectively, is apparently not achieved. Regardless, only vaccination coverage higher than 95% permits good herd immunity for Me [16], and for optimal herd immunity, everyone should be vaccinated [17].

In the most recent position paper on Me, the WHO [18] concluded that “although vaccine-induced antibody concentrations decline over time and may become undetectable, immunological memory persists and, following exposure to measles virus, most people who have been vaccinated produce a protective immune response”. Nevertheless, Me cases may also occur in twice-vaccinated subjects [19,20]. Waning antibodies and differences in virus genotype of the vaccine and of circulating strain may be a possible explanation [21]; in contrast, older studies stated that cellular immunity induced by the Me vaccine probably protects vaccinated subjects [22] despite trace amounts of circulating antibodies [23]. Further, waning immunity may occur due to a lack of boosting from circulating wild-type virus [24]. Moreover, the waning of both the concentration and the avidity of antibodies might contribute to Me infection in twice-MMR-vaccinated subjects [25].

The position of WHO on measles is clear and does not propose booster doses over time after the two-dose cycle. This is justifiable in the general population, but what about for HCWs? The issue, at least in Italy, is debated. Quite recently [26], in subjects vaccinated with two doses during childhood, one booster dose guaranteed a seroconversion of 74% and a second of 93%, without significant adverse effects. However, is it ethically acceptable to suggest a booster dose not deemed necessary by international authorities? Nevertheless, a third dose of vaccine is also suggested in twice-vaccinated subjects, in particular in HCWs [27]. The outbreak of Me in Italy in 2017 caused 5407 cases of illness, 4 of which were followed by death. The vast majority of cases of disease occurred in unvaccinated patients, but 7.2% were vaccinated once and 1.6% twice; the low vaccination coverage of HCWs was also the cause of 334 illnesses among these individuals [28].

The present research has two weaknesses: (1) without knowing the type of vaccine (especially for the MMR vaccine) used, the evaluation of the 1991–1995 cohort is speculative; and (2) the antibody measurement was done with an EIA method, but the measurement of neutralizing antibodies might be better correlated with protection [29,30]. On the other hand, the strengths of the study lie in the size of the sample, the use of vaccination certificates (although they lack the information noted above), and the completeness of the health documentation.

## 5. Conclusions

In conclusion, the question posed in the title can be answered affirmative; we will have a new generation of HCWs covered by MMR vaccination. The 95% herd immunity for Me will probably be overcome, and a more complex evaluation regarding the immune coverage, which is optimal for Ru (probably a single dose would be enough) but not for Mu and Me, will be achieved. What worries us most is Me, given the possible immediate and delayed complications as a consequence of viral infection. On the other hand, accordingly to the WHO [18] a full cycle of MMR in infancy should be enough to ensure adequate coverage even in the event of the disappearance of circulating antibodies. Finally, an open problem is if and when a subject must be declared a non-responder.

## Figures and Tables

**Figure 1 vaccines-08-00104-f001:**
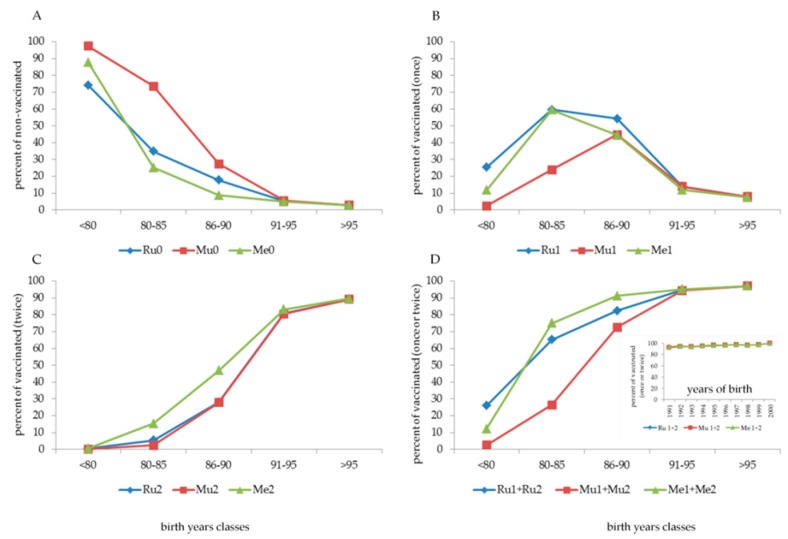
Percentage of subjects not vaccinated (**panel A**) or vaccinated with one (**panel B**), two (**panel C**) or one+two (**panel D**) doses according to date of birth classes. In the box at the bottom of panel D, the percentage of vaccinated subjects born between 1991 and 2000 with one+two doses is presented.

**Figure 2 vaccines-08-00104-f002:**
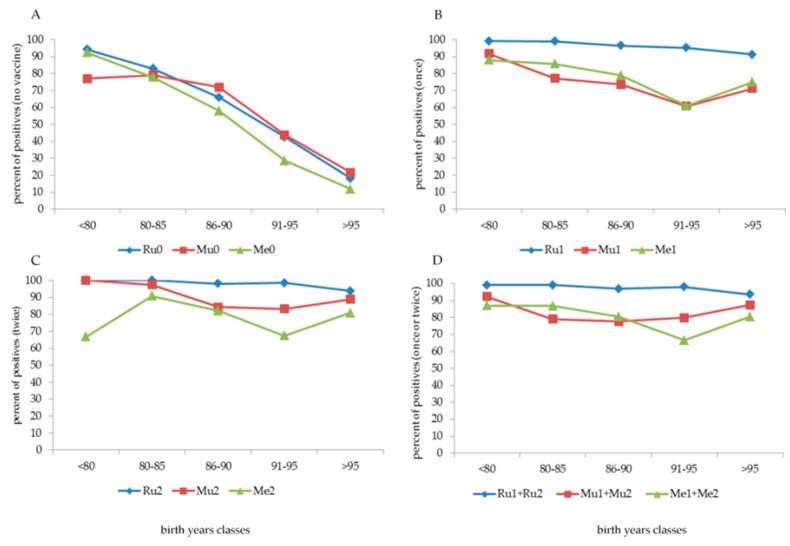
Percentage of positive antibodies against Ru, Mu and Ru in subjects who were not vaccinated (**panel A**) and those who were vaccinated with one (**panel B**), two (**panel C**) or one+two (**panel D**) doses according to date of birth classes.

**Figure 3 vaccines-08-00104-f003:**
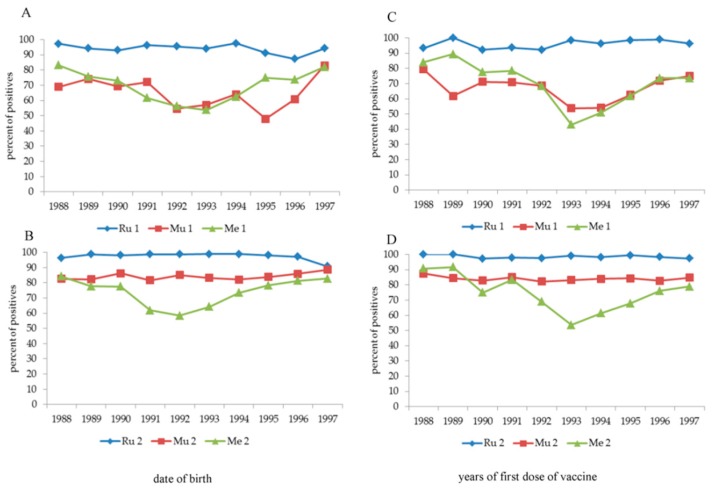
Percentage of positive antibodies against Ru, Mu and Ru after one or two doses of vaccine according years of the date of birth (**panel A**,**B**) and the 1st dose of vaccine (**panel C**,**D**) in students born between 1988 and 1997.

**Table 1 vaccines-08-00104-t001:** Type of vaccination against Ru, Mu and Me (percentage in parentheses).

**Date of Birth**	**N.**	**Ru0**	**Ru1**	**MMR1**	**Ru+Ru**	**Ru+MMR**	**MMR2**
before 1980	495	367 (74.1)	115 (23.2)	10 (2.0)	0	2 (0.4)	1 (0.2)
1980–1985	1544	538 (34.8)	626 (40.5)	295 (19.1)	3 (0.2)	51 (3.3)	31 (2.0)
1986–1990	2685	474 (17.7)	285 (10.6)	1171 (43.6)	0	29 (1.1)	726 (27.0)
1991–1995	3892	201 (5.2)	15 (0.4)	522 (13.4)	0	24 (0.6)	3126 (80.3)
after 1995	2037	60 (2.9)	1 (0.05)	160 (7.9)	0	2 (0.1)	1814 (89.1)
**Date of Birth**	**N.**	**Mu0**	**Mu1**	**MMR1**	**Mu+Mu**	**Mu+MMR**	**MMR2**
before 1980	495	482 (97.4)	0	12 (2.4)	0	0	1 (0.2)
1980–1985	1544	1137 (73.6)	30 (1.9)	339 (22.0)	0	7 (0.5)	31 (2.0)
1986–1990	2685	736 (27.4)	23 (0.9)	1180 (43.9)	0	20 (0.7)	726 (27.0)
1991–1995	3892	219 (5.6)	1 (0.03)	543 (14.0)	0	3 (0.08)	3126 (80.3)
after 1995	2037	60 (2.9)	1 (0.05)	162 (8.0)	0	0	1814 (89.1)
**Date of Birth**	**N.**	**Me0**	**Me1**	**MMR1**	**Me+Me**	**Me+MMR**	**MMR2**
before 1980	495	434 (87.7)	48 (9.7)	10 (2.0)	0	2 (0.4)	1 (0.2)
1980–1985	1544	388 (25.1)	760 (49.2)	156 (10.1)	19 (1.2)	188 (12.2)	33 (2.1)
1986–1990	2685	234 (8.7)	505 (18.8)	690 (25.7)	20 (0.7)	510 (19.0)	726 (27.0)
1991–1995	3892	189 (4.9)	30 (0.8)	435 (11.2)	1 (0.03)	111 (20.9)	3126 (80.3)
after 1995	2037	59 (2.9)	2 (0.1)	150 (7.4)	0	13 (0.6)	1814 (89.1)

Legend: Ru/Mu/Me0 = not vaccinated; Ru/Mu/Me1 or Ru+Ru; Mu+Mu; Me+Me = vaccinated with specific vaccine alone; MMR1 = vaccinated once with trivalent vaccine; Ru/Mu/Me+MMR = vaccinated with specific vaccine alone and MMR; MMR2 = vaccinated twice with trivalent vaccine.

**Table vaccines-08-00104-t002a:** 

Ru	b	r	t	p
intercept	3.25218		22.77316	< 0.0001
**date of birth**	**−0.000021**	**−0.093403**	**−5.306036**	**< 0.0001**
**date of vaccine**	**0.000012**	**0.039619**	**2.242623**	**= 0.025**
**interval**	**−0.00001**	**−0.038574**	**−2.183368**	**= 0.0291**

Ru Ab one dose = 3.25218 −0.000021 date of birth +0.000012 date of vaccine −0.00001 interval.

**Table vaccines-08-00104-t002b:** 

Mu	b	r	t	p
intercept	4.330541		7.786166	< 0.0001
**date of birth**	**−0.00008**	**−0.088679**	**−4.257648**	**< 0.0001**
date of vaccine	0.000026	0.024039	1.149929	= 0.2503
interval	−0.000016	−0.015281	−0.730871	= 0.4649

Mu Ab one dose = 4.330541 −0.00008 date of birth +0.000026 date of vaccine −0.000016 interval.

**Table vaccines-08-00104-t002c:** 

Me	b	r	t	p
intercept	5.806215		15.709522	< 0.0001
**date of birth**	**−0.000058**	**−0.083567**	**−4.424757**	**< 0.0001**
date of vaccine	−0.000027	−0.032669	−1.724637	= 0.0847
**interval**	**−0.000054**	**−0.070944**	**−3.75272**	**= 0.0002**

Me Ab one dose = 5.806215 −0.000058 date of birth −0.000027 date of vaccine −0.000054 interval.

**Table vaccines-08-00104-t003a:** 

Ru	b	r	t	p
intercept	3.493779		25.100325	< 0.0001
date of birth	−0.000006	−0.0122	−0.929521	= 0.3527
date of 1st dose	+6.60E-07	0.002247	0.171188	= 0.8641
date of 2nd dose	−0.000007	−0.012479	−0.95077	= 0.3418
**interval since the last dose**	**−0.000023**	**−0.048249**	**−3.680109**	**= 0.0002**

Ru Ab two doses = 3.493779 −0.000006 date of birth +6.60E-07 date 1st dose −0.000007 date 2nd dose −0.000023 interval since the last dose.

**Table vaccines-08-00104-t003b:** 

Mu	b	r	t	p
intercept	1.432912		3.692206	= 0.0002
date of birth	−0.000018	−0.012968	−0.981099	= 0.3266
date of 1st dose	−0.000009	−0.010251	−0.77555	= 0.438
**date of 2nd dose**	**0.000057**	**0.037773**	**2.859599**	**= 0.0043**
interval since the last dose	0.000029	0.022088	1.67139	= 0.0947

Mu Ab two doses = 1.432912 −0.000018 date of birth −0.000009 date 1st dose +0.000057 date 2nd dose +0.000029 interval since the last dose.

**Table vaccines-08-00104-t003c:** 

Me	b	r	t	p
intercept	4.801932		13.227424	< 0.0001
date of birth	−0.000002	−0.001449	−0.117342	= 0.9066
**date of 1st dose**	**0.000092**	**0.083703**	**6.80124**	**< 0.0001**
**date of 2nd dose**	**−0.000133**	**−0.08061**	**−6.54827**	**< 0.0001**
**interval since the last dose**	**−0.000067**	**−0.047604**	**−3.858866**	**= 0.0001**

Me Ab two doses = 4.801932 −0.000002 date of birth +0.000092 date 1st dose −0.000133 date 2nd dose −0.000067 interval since the last dose.

**Table 4 vaccines-08-00104-t004:** Age of vaccination of students vaccinated once.

Date of Birth	Vaccine Type	N.	Mean±SD	Median	Range
before 1980	Ru1	125	11.6 ± 5.0	10.9	< 1–37
	Mu1	12	21.6 ± 11.9	23.0	< 1–37
	Me1	58	8.7 ± 8.7	5.5	1–37
1980–1985	Ru1	921	10.6 ± 2.7	10.9	< 1–26
	Mu1	369	10.2 ± 4.8	11.0	< 1–24
	Me1	918	4.0 ± 4.3	2.0	< 1–25
1986–1990	Ru1	1456	8.1 ± 4.7	11.0	< 1–24
	Mu1	1203	7.4 ± 4.9	10.5	< 1–24
	Me1	1195	3.6 ± 3.6	1.8	< 1–24
1991–1995	Ru1	540	6.1 ± 5.4	2.4	< 1–21
	Mu1	544	6.0 ± 5.4	2.3	< 1–21
	Me1	465	4.4 ± 4.7	1.7	< 1–21
after 1995	Ru1	161	3.7 ± 4.3	1.5	1–20
	Mu1	163	3.6 ± 4.2	1.5	1–20
	Me1	152	3.2 ± 4.0	1.5	1–20

**Table 5 vaccines-08-00104-t005:** Age of vaccination of students vaccinated twice.

			Age at 1st Dose			Age at 2nd Dose		
Date of Birth	Vaccine Type	N.	Mean±SD	Median	Range	Mean±Sd	Median	Range
1980–1985	Ru2	85	6.4 ± 4.7	7.2	< 1–21	14.0 ± 4.5	11.9	10–27
	Mu2	38	3.4 ± 4.3	1.9	1–21	12.7 ± 4.3	11.0	10–27
	Me2	238	2.6 ± 2.1	1.7	< 1–21	11.9 ± 3.1	11.4	< 1–21
1986–1990	Ru2	755	2.3 ± 2.3	1.5	1–17	12.3 ± 2.3	11.6	1–24
	Mu2	746	2.3 ± 2.2	1.5	1–17	12.3 ± 2.2	11.5	1–21
	Me2	1256	2.0 ± 1.7	1.5	< 1–17	12.0 ± 2.1	11.5	1–23
1991–1995	Ru2	3150	2.1 ± 1.9	1.5	< 1–19	11.7 ± 2.0	11.7	< 1–23
	Mu2	3129	2.0 ± 1.8	1.5	< 1–19	11.7 ± 2.0	11.7	< 1–23
	Me2	3238	2.0 ± 1.8	1.5	< 1–19	11.7 ± 2.0	11.7	< 1–23
after 1995	Ru2	1816	1.7 ± 1.5	1.4	< 1–19	8.2 ± 2.6	7.7	1–20
	Mu2	1814	1.7 ± 1.5	1.4	< 1–19	8.2 ± 2.6	7.7	1–20
	Me2	1826	1.7 ± 1.5	1.4	< 1–19	8.2 ± 2.6	7.7	1–20

Note: Age classes born before 1980 were excluded because they were too limited in number.

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
