# Peer review of "Will We Have a Cohort of Healthcare Workers Full Vaccinated against Measles, Mumps, and Rubella?"

_vaccines, 2020, doi:10.3390/vaccines8010104_

Round 1

Reviewer 1 Report

The manuscript by Trevisan et al. addresses the difficult problem of vaccine/immumologcial coverage in healthcare workers (HCWs). The authors sampled a large cohort of students belonging to the Medical School of Padua University susceptible to contract and/or to transmit the infection owing to their future job, were for MMR vaccination and immune coverage. The overall aim was to evaluate whether, in the near future, it will be possible to have a new generation of HCWs fully vaccinated (and immunized) against measles, mumps and rubella.

Overall the manuscript is well written and while based on Italian students will be of interest to epidemiologist world-wide interested the in the immunological status of HCWs.

Comments

Major:

Tables 4 and 5. All results should be presented in the results section. Adding addition results the discussion section is discouraged. I would recommend moving Tables 4 and 5 and associated text to the results section.

Minor:

Line 40. The abstract should stand alone. In some instances it may be the only part of the paper available. I would recommend removing the question at the end OR provide the answer.

Line 49. Please provide further information/background on the “Pisa card”.

Line 102. The common acronym for enzyme-linked immunosorbent assay is ELSIA.

Lines 105-106 and 115-117. The authors cite the CDC recommendations that equivocal were processed as negative. But later state that they categorized negative titers as 1, equivocal as 2 and positive as 3. This should be better explained.

Table 2 and 3. b, r, t, and p should be defined.

Author Response

The manuscript by Trevisan et al. addresses the difficult problem of vaccine/immumologcial coverage in healthcare workers (HCWs). The authors sampled a large cohort of students belonging to the Medical School of Padua University susceptible to contract and/or to transmit the infection owing to their future job, were for MMR vaccination and immune coverage. The overall aim was to evaluate whether, in the near future, it will be possible to have a new generation of HCWs fully vaccinated (and immunized) against measles, mumps and rubella.

Overall the manuscript is well written and while based on Italian students will be of interest to epidemiologist world-wide interested the in the immunological status of HCWs.

Reply: The authors thank the reviewer for the sincere appreciation of the work done.

Comments

Major:

Tables 4 and 5. All results should be presented in the results section. Adding addition results the discussion section is discouraged. I would recommend moving Tables 4 and 5 and associated text to the results section.

Reply: Table 4 and 5 have been moved to the results section

Minor:

Line 40. The abstract should stand alone. In some instances it may be the only part of the paper available. I would recommend removing the question at the end OR provide the answer.

Reply: the sentence at the end of abstract was removed

Line 49. Please provide further information/background on the “Pisa card”.

Reply: Pisa card is an initiative launched by Pier Luigi Lopalco during the conference indicated in the text and first shared with me (Andrea Trevisan) who at the time was the national coordinator for the Italian society of occupational medicine in the section dedicated to health workers and then shared with other scientific societies that strongly declared that health workers had to be particularly stimulated to perform, if they had not done the 7 vaccinations also indicated by the PNPV 2017-2019. I remember that at that time Italy was in the midst of the measles epidemic.

Line 102. The common acronym for enzyme-linked immunosorbent assay is ELSIA.

Reply: the acronym was changed accordingly

Lines 105-106 and 115-117. The authors cite the CDC recommendations that equivocal were processed as negative. But later state that they categorized negative titers as 1, equivocal as 2 and positive as 3. This should be better explained.

Reply: equivocal results were categorized to a better evaluation of multiple linear analysis. On the other hand, to evaluate seroprevalence of positive antibodies equivocal results the equivocal results were considered, as described, among the negatives.

Table 2 and 3. b, r, t, and p should be defined.

Reply: b=slope, r=regression coefficient, t=t test, p=significance.

Reviewer 2 Report

This is an excellent study for the appropriateness of vaccination and immune coverage in health care workers.

The authors have performed an excellent review in this population dating back to 1980. Their conclusion and statistical analysis were appropriate and solid.

A minor recommendation is to be consistent in the graphs. Graphs 1 and 2 do not have the legend "Year of Birth" that is present in Figure 3. Please make these changes to Figures 1 and 2.

Author Response

Comments and Suggestions for Authors

This is an excellent study for the appropriateness of vaccination and immune coverage in health care workers.

The authors have performed an excellent review in this population dating back to 1980. Their conclusion and statistical analysis were appropriate and solid.

Reply: the authors thank the reviewer for their appreciation of their work.

A minor recommendation is to be consistent in the graphs. Graphs 1 and 2 do not have the legend "Year of Birth" that is present in Figure 3. Please make these changes to Figures 1 and 2.

Reply: as required, changes have been made to figures 1 and 2